# Analysis of Gait Kinematics in Smart Walker-Assisted Locomotion in Immersive Virtual Reality Scenario

**DOI:** 10.3390/s24175534

**Published:** 2024-08-27

**Authors:** Matheus Loureiro, Arlindo Elias, Fabiana Machado, Marcio Bezerra, Carla Zimerer, Ricardo Mello, Anselmo Frizera

**Affiliations:** 1Graduate Program in Electrical Engineering, Federal University of Espírito Santo, Vitória 29075-910, ES, Brazil; matheus.loureiro@edu.ufes.br (M.L.); marcio.bezerra@edu.ufes.br (M.B.); carla.zimerer@ufes.br (C.Z.); ricardo.c.mello@ufes.br (R.M.); 2Graduate Program in Physiotherapy, Estacio de Sa University, Vitória 29092-095, ES, Brazil; arlindo.neto@estacio.br; 3Graduate Program in Informatics, Federal University of Espírito Santo, Vitória 29075-910, ES, Brazil; fabiana.machado@ufes.br

**Keywords:** gait analysis, motion capture, rehabilitation robotics, smart walker, virtual reality

## Abstract

The decline in neuromusculoskeletal capabilities of older adults can affect motor control, independence, and locomotion. Because the elderly population is increasing worldwide, assisting independent mobility and improving rehabilitation therapies has become a priority. The combination of rehabilitation robotic devices and virtual reality (VR) tools can be used in gait training to improve clinical outcomes, motivation, and treatment adherence. Nevertheless, VR tools may be associated with cybersickness and changes in gait kinematics. This paper analyzes the gait parameters of fourteen elderly participants across three experimental tasks: free walking (FW), smart walker-assisted gait (AW), and smart walker-assisted gait combined with VR assistance (VRAW). The kinematic parameters of both lower limbs were captured by a 3D wearable motion capture system. This research aims at assessing the kinematic adaptations when using a smart walker and how the integration between this robotic device and the VR tool can influence such adaptations. Additionally, cybersickness symptoms were investigated using a questionnaire for virtual rehabilitation systems after the VRAW task. The experimental data indicate significant differences between FW and both AW and VRAW. Specifically, there was an overall reduction in sagittal motion of 16%, 25%, and 38% in the hip, knee, and ankle, respectively, for both AW and VRAW compared to FW. However, no significant differences between the AW and VRAW kinematic parameters and no adverse symptoms related to VR were identified. These results indicate that VR technology can be used in walker-assisted gait rehabilitation without compromising kinematic performance and presenting potential benefits related to motivation and treatment adherence.

## 1. Introduction

The proportion of elderly people in the world has been increasing over the years, and the pace of aging is expected to intensify in the coming decades [1]. This population profile requires a large number of healthcare professionals and institutions to maintain well-being and quality of life status, which contributes to higher health-related direct and indirect costs [2].

The World Health Organization (WHO) describes healthy aging as the development and maintenance of capabilities that enable the elderly to be and do their activities independently [3]. Nevertheless, the inherent decline in physical and cognitive capabilities faced by aging individuals can affect their independence levels [4]. Musculoskeletal dysfunctions are important consequences of this decline and are manifested as the loss of muscle strength, balance disorders, and gait impairments [5]. The evolution of musculoskeletal problems leads to mobility impairments that affect an individual’s ability to move freely and perform daily tasks [6]. Furthermore, it increases the risk of falls, which is the most common cause of accidental death in the elderly [1,5].

Gait and balance disorders are also common in people affected by neurological problems, such as cerebral palsy, multiple sclerosis, parkinson, and stroke [5]. Rehabilitation plays an important role in these people’s lives by improving independence and confidence during motion tasks. The WHO emphasizes that rehabilitation should be available for the entire world population and all ages [7].

Rehabilitation typically involves repetitive tasks that are performed during a certain timeline, which can be very long depending on the subject’s condition [8]. This process is not linear and can be further complicated by the presence of pain, frustration related to not accomplishing partial goals, and failure to notice improvement in daily tasks. These factors can have a significant impact on the rehabilitation’s dropout rates [8,9]. Recently, technologies, such as robot-assisted gait training, virtual reality systems, and wearable sensors have been introduced into the biomechanical and rehabilitation fields integrated into training to enhance motor recovery, increase motivation, and improve treatment reports and the patient’s adherence [10].

Virtual reality (VR) is an emerging technology that has been explored to improve such goals and develop scenarios to investigate balance issues, gait analysis, and assistive device developments [11,12]. VR technologies can be used to create personalized environments according to the patient’s needs in gait rehabilitation [11,13]. Several studies have compared the impact of VR strategies to traditional methods of balance and gait rehabilitation [11].

VR setups have been utilized not only for gait rehabilitation but also for the rehabilitation of upper and lower limbs in general [14,15]. VR scenarios can be combined with game interfaces, particularly serious games, to enhance user engagement and motivation while performing activities [15,16]. For example, in [17], a VR game was developed for hand rehabilitation of subacute stroke patients, incorporating exercises like grasping and hitting obstacles within a game setting. Additionally, in [16], volunteers using a treadmill were motivated to walk through a VR forest accompanied by a virtual dog, which aimed to increase motivation during gait rehabilitation. At the end of the walk, participants reported a reduced physical demand while immersed in the game.

Despite suggestions that VR can provide rehabilitation that is more effective and motivated than traditional rehabilitation, little evidence exists to date to support these claims [11,15,18,19]. The use of VR can cause some adverse symptoms, such as nausea, disorientation, headache, and eye strain. These symptoms are called cybersickness and can occur after a VR session [20]. Besides cybersickness, VR systems can also affect gait biomechanical parameters, such as reductions in walk speed, stride length, or cadence, when compared with a traditional rehabilitation system [21,22,23].

For example, the works presented in [21,22] compared overground walking in the physical world and in immersive VR. The results showed that the participants demonstrated small changes in some parameters of walk distance, and stride and step length, but some significant differences were found in ankle plantar flexion, cadence, and walking speed. The authors suggested that these changes could be associated with a more conservative and cautious walk pattern due to instability induced by the VR environment [22].

Immersive VR walks can also be combined with robotic assistive devices. These devices are divided into stationary systems, where the robot is fixed in a structure, for example, a treadmill, or overground walking systems, where the user can move freely in a mobile base robot, such as an exoskeleton or smart walker (SW) [24,25,26]. There are several studies on gait rehabilitation with VR on instrumented treadmills, because stationary systems had fewer requisites related to actuators and maneuverability to be developed with VR than overground walking systems [27,28,29].

However, it is important to evaluate VR scenarios with overground walking systems, as this helps the user practice a more natural gait without restrictions and also can improve the outcomes of rehabilitation programs [25,30]. SW stands out in overground walking robots because of their potential to improve balance and provide body weight support during a walk [31]. Additionally, such devices can include monitoring of biomechanical parameters, fall detection algorithms, obstacle avoidance strategies, and guidance to follow specific paths [31]. These features can be used in the rehabilitation of a wide spectrum of neurological and cognitive deficits [24,32,33,34].

Over the past decade, our research group has been developing research in the rehabilitation robotic field, focusing on applications for SW. We developed strategies to enhance human–robot interaction whilst empowering safe navigation respecting the user’s residual capabilities [35,36,37,38,39]. Furthermore, we integrated our SW with cloud computing systems to execute tasks with high computational cost [40]. Recently, our works have focused on integrating SW in VR to develop interactive scenarios for gait training and rehabilitation, combining physical and virtual environments [32,41,42]. The results indicate positive acceptance of the VR system, as shown by questionnaire feedback, and task performance improvements, mainly in execution time, when gait tasks were performed using VR and SW. Nevertheless, these works did not include biomechanical parameters.

The monitoring of human biomechanical parameters during gait is an important tool to diagnose pathologies and evaluate treatments [43]. The recent advancements in the development of wearable systems based on Inertial Measurement Units (IMUs) to track three-dimensional kinematic variables allowed for monitoring these human parameters in real-world scenarios (out of the laboratory) in several fields, including rehabilitation and sports [44]. The combination of this wearable system with SW or other assistive devices can be used as an additional tool to study the kinematics effects during a walk with these devices [34]. However, the clinical applications of such device combinations are scarce in the literature and require investigation.

This paper investigates the gait kinematic parameters of elderly individuals across three experimental tasks: normal free walking (FW), smart walker-assisted gait (AW), and smart walker-assisted gait plus VR assistance (VRAW). The FW task was designed to capture the standard walking patterns of each individual, which were then compared with the patterns observed in the AW and VRAW tasks. The goal of this paper is to evaluate if the differences between FW and AW are the same as those between FW and VRAW and how AW and VRAW differ from each other to assess the effects of VR in gait-assisted tasks. The results can help in understanding the kinematic adaptations to the use of an SW and how the VR environment can influence such adaptations and can ultimately provide an integration strategy between different gait assistance technologies.

## 2. Materials and Methods

### 2.1. Participants

A total of 14 random elderly individuals (5 men and 9 women) from the local community were enrolled in the experiment. These volunteers were recruited from an extension program at our university that offers recreational activities to the elderly population and an external exercise guidance center within the community. The sample size calculation was based on a repeated-measures ANOVA, within factors, of one group of participants and three measurements. The level of significance and power were set to 0.05 and 0.80, respectively, and the expected effect size was moderate. The calculations were performed in G.Power (version 3.1.9.2). The participants presented a mean age of 66.3 (±3.9) years, mean weight of 71.43 (±10.8) kg, and mean height of 1.66 (±0.09) m. The inclusion criteria consisted of ages above 60 years old and having enough physical condition for the walk test and training with the smart walker device. Subjects were excluded in the case of additional comorbidities that prevented the walking tests, previous experience with VR technology, being a smoker or under thermogenic supplements or medications use, and scoring above 24 points on the Mini-Mental test to ensure proper cognitive function [45]. This information was obtained by conducting an online interview before receiving the participants in our laboratory. During this interview, we performed an anamnesis to collect comprehensive details about each participant’s medical history, including previous and current illnesses, disorders, living conditions, and potential risk factors. Additionally, we administered the Mini-Mental State Examination to assess cognitive function. The participants were also encouraged to maintain their usual diet and medication use throughout the experiments, and they were also informed about the potential risks and discomforts associated with the research procedures and were required to voluntarily sign a Consent Form. This experiment was approved by the Research Ethics Committee of the Federal University of Espírito Santo (registration number 6.294.101).

### 2.2. Materials

The materials used in this paper were the UFES vWalker, an SW developed by our group [32], a VR headset (Oculus Quest 2, Meta Quest, Menlo Park, CA, USA), and a 3D motion capture IMU-based system (MVN Awinda, Movella, NV, USA). These materials were divided into five subsystems: Odometry and Control (OC), Human–Robot-Environment Interaction (HREI), Human–Robot Interaction (HRI), Motion Capture (MC), and Virtual Reality Integration (VRI). The first three subsystems were related to the UFES vWalker, and the last two were associated with the VR headset and 3D motion capture system, respectively. All the materials and subsystems are shown in Figure 1.

### 2.3. UFES vWalker

From the subsystems of the UFES vWalker, the OC subsystem is responsible for odometry and localization using an IMU (BNO055 9-DOF BOSCH, Gerlingen, Germany) and two encoders (H1 series—US Digital, Vancouver, WA, USA). It also facilitates propulsion and stability through two motorized wheels and two caster wheels. The HREI subsystem is composed of a Light Detection and Ranging (LiDAR) system (URG-04LX Hokuyo, Osaka, Japan) used in front of the SW for obstacle identification.

The HRI subsystem is composed of another LiDAR system (RPLIDAR A3 SLAMTEC, Shanghai, China) used in front of the user’s legs to track the distance between the user and the device, using a leg clustering technique proposed in [37]. Another component of HRI is the two triaxial force sensors (MTA400 FUTEK, Irvine, CA, USA) responsible for capturing the user’s movement intention and converting it into linear $\nu_{c}\left(t\right)$ and angular velocity $w_{c}\left(t\right)$ commands with the admittance controller proposed in [35]. The controller is shown in Equations (Equation 1)–(Equation 4).
(1)$$F\left(t\right)=-\left(\frac{{F_{L}}_{Y}\left(t\right)+{F_{R}}_{Y}\left(t\right)}{2}\right),$$
(2)$$\tau \left(t\right)=-\left(\frac{{F_{L}}_{Y}\left(t\right)-{F_{R}}_{Y}\left(t\right)}{2}\right)d.$$
(3)$$\nu_{c}\left(t\right)=\frac{F\left(t\right)-m_{\nu }\dot{\nu }\left(t\right)}{d_{\nu }},$$
(4)$$w_{c}\left(t\right)=\frac{\tau \left(t\right)-m_{\omega }\dot{\omega }\left(t\right)}{d_{\omega }}.$$

${F_{L}}_{Y}$ and ${F_{R}}_{Y}$ are the forces on the left and right arm, respectively. These forces are used to calculate the force forward F(t) and torque τ(t) in Equations (Equation 1) and (Equation 2), where *d* is the distance between the left and right sensors. Then, it generates the UFES vWalker linear $v_{c}\left(t\right)$ and angular reference velocities $w_{c}\left(t\right)$ in Equations (Equation 3) and (Equation 4), respectively. The constants $d_{\nu }$ and $d_{\omega }$ are damping parameters, $m_{\nu }$ and $m_{w}$ are virtual masses, and $\dot{\nu }\left(t\right)$ and $\dot{\omega }\left(t\right)$ represent the linear and angular acceleration.

The admittance controller is used in the UFES vWalker to promote a more natural interaction between the user and robot [46]. With this controller, it is possible to modulate the SW inertia, emulating different haptic levels to the user and controlling the device through the force sensors.

The data from the sensors mentioned above are captured by a microcontroller (STM32) that is responsible for the low-level control. This information is sent to a minicomputer (OptiPlex Micro, 16 GB RAM) (DELL, Austin, TX, USA) with a middleware Robot Operating System (ROS), used to control the UFES vWalker.

### 2.4. The 3D Motion Capture System

The participant’s gait information is captured using a wireless 3D motion capture IMU-based system. This portable equipment has been previously validated for extracting gait information [44,47]. In this paper, this equipment comprised the MC subsystem. Seven IMUs were attached by straps in one in each foot, lower leg, and upper leg and one in the pelvis of the participants, as shown in Figure 1. This information is combined with the individual’s anthropometric measurements to obtain the 3D gait information of the lower limbs in the developer software (MVN Analyze—Version: 2024.1, MVN Awinda, Movella, NV, USA).

### 2.5. Immersive VR Scenario

The VRI subsystem is responsible for the immersion of the UFES vWalker and the participants in the VR scenario, which was made using the software Unity, a game engine that enables the development of games and other applications. Using the VR headset, as shown in Figure 1, the participants are able to see the user interface. In this interface, the participant sees a star, and each time they collect a star with the UFES vWalker, the score increases in the left corner of the screen, and a new star appears in front of them. The stars were distributed to guide the participant in a straight line, and the black lines on the ground marked the boundaries of the physical world, considering that the experiments were conducted in a 60 m × 5 m hallway.

The user interface is also a feedback interface used to assist participants in maintaining safe navigation and enhancing their presence in the VR scenario. The feet in the middle represent the distance from the user’s leg to the device. The dotted lines are the limits that guarantee a safe distance to the SW. The minimum distance was 0.25 m and the maximum distance was 0.75 m. Each sidebar represents the downward force applied in the force sensor: green bars indicate a correct body weight discharge, while red bars indicate that the user is poorly positioned on the UFES vWalker. A downward force above 5% of the participant’s weight was considered a correct discharge. If the users do not meet the specified distance or force values, the UFES vWalker remains stopped to avoid risk to the participants.

Figure 1 on the right side also shows the digital twin of the UFES vWalker. Every movement of the UFES vWalker in the physical world is replicated in the VR scenario using odometry messages from the OC subsystem through a ROS protocol (ROS#). The motion capture data are also reproduced in the VR scenario using a Unity plugin in MVN Analyze. This information is used to monitor the user during the experiments.

### 2.6. Experimental Protocol

The participants were randomly divided into two equally distributed groups. Each group participated in three days of experiments, separated by 48 h each day. During the experiments, the participants completed three different gait scenario tasks. The first task was the free walk (FW): a 10-Meter Walk Test (10 MWT) without the UFES vWalker and the VR headset. The second task was the UFES vWalker-Assisted Walk (AW): a straight-line walk for 90 s with the UFES vWalker. The last task was the UFES vWalker Virtual Reality-Assisted Walk (VRAW): a straight-line walk for 90 s with the UFES vWalker and the VR headset. In all the tasks, the participants used the wearable motion capture system.

The FW was designed to extract the standard walking patterns of each individual. These patterns were then compared with the AW and VRAW. It was expected that a walk assisted by SW (whether with or without VR) would differ from a free walk due to body weight support and other characteristics related to the SW. Nonetheless, the goal of this paper is to evaluate if the differences between FW and AW are the same as those between FW and VRAW and how AW and VRAW differ from each other to assess the effects of VR in gait-assisted tasks.

On the first day, the UFES vWalker, the VR headset, and the motion capture system were introduced to the participants, and they performed a familiarization test consisting of one FW trial, one AW trial, and one VRAW trial. The data from these initial tests were not used in the analysis. If any volunteers were unable to perform one of the tests due to physical limitations, they would be excluded from the research. On the second day, half of the participants performed one FW and three AW tasks, while the other half performed one FW and three VRAW tasks. On the third day, the groups swapped their tasks from day two. The goal of this group distribution was to ensure that half of the participants started the second day with AW tasks and the other half with VRAW tasks.

This experimental protocol was designed as a gait training study using the UFES vWalker to evaluate the effects of incorporating a VR headset on gait parameters and cybersickness symptoms. Based on the results obtained, future protocols could be adapted to rehabilitation settings to explore the practical applications of the UFES vWalker in gait rehabilitation contexts.

### 2.7. Variables

This paper investigated two groups of kinematic variables of the gait cycle across three gait scenario tasks during a 10 MWT. The range of 10 m in a straight line was chosen because it is a tool standard to measure the functional capacity of people in gait analysis [48]. For the AW and VRAW tasks, the 10 MWT was selected from the middle portion of the 90 s walk, during the period when the participants completed this distance. The first group of variables included the overall spatiotemporal parameters of the walking cycle, and these variables were the following:Stride length (meters): the mean of the distance between two consecutive heel strikes of the same foot in the 10 MWT.Stride number: the number of steps in the 10 MWT.Gait speed (meters per second): the mean walk velocity in the 10 MWT.Cadence (steps per second): the mean number of steps per second in the 10 MWT.Stance phase (seconds): the mean time in the stance phase in each gait cycle in the 10 MWT.Swing phase (seconds): the mean time in the swing phase in each gait cycle in the 10 MWT.Time (seconds): the time to complete the 10 MWT.

The second group of variables consisted of lower limb joint angles of the hip, knee, and ankle, extracted from specific instants (mainly extreme points) of each joint cycle waveform in the sagittal, coronal, and transverse planes. The parameters are based on the work by [48]. These parameters are summarized in Table 1. These parameters are also shown in a gait cycle from a random volunteer during the FW in Figure 2 for the hip, knee, and ankle, respectively. A 3D kinematic gait analysis relies on extensive data that can be complex to interpret visually [49]. To facilitate the evaluation of the volunteer’s gait patterns, the gait point segmentation method proposed in [48] is used, enhancing the clarity of the analysis [49].

All these variables were extracted from MVN Analyze, which exports a specific file extension (.mvnx) with all the 3D gait data information. These files were imported in MATLAB (R2024a), where the heel and toe strikes of the foot were identified to extract the variables from each gait cycle during the 10 MWT.

Before starting the VRAW task, the volunteers completed the Simulator Sickness Questionnaire (SSQ) to identify any presence of motion sickness symptoms when using VR systems [50]. Volunteers who exhibited any of these symptoms before the experiments were excluded from the research. After the VRAW task, the volunteers were invited to fill out the SEQ (Suitability Evaluation Questionnaire for Virtual Rehabilitation Systems) at the end of the experiment [51]. This questionnaire is composed of fourteen questions with a 5-point Likert Scale, resulting in a score range from 0 to 65. The questions evaluate the perception of the participants about usability, acceptance, and security and detect frequent VR issues (cybersickness symptoms such as nausea, eye discomfort, and disorientation) after using VR rehabilitation systems.

### 2.8. Statistical Analysis

The distributions of the spatiotemporal and kinematic variables were analyzed graphically using histograms and statistically with the Shapiro–Wilk test to check for significant deviations from normality. The general features of each variable were analyzed by descriptive statistics. A repeated-measures one-way ANOVA was conducted to compare each kinematic variable across the different gait tasks (FW, AW, and VRAW). In the repeated-measures design, sphericity is an important assumption that refers to the equality of variances of the differences between groups. Mauchly’s test was used to address sphericity issues in each comparison and if the result was significant, then Greenhouse–Geiser correction was applied to adjust the degrees of freedom and provide a more reliable F-test [52]. A *p*-value of less than 0.05 indicated a statistically significant difference across the three gait tasks. The Bonferroni post hoc adjustment was used to determine specific between-group differences, with a *p*-value of less than 0.05 indicating significance. In the event of normality violations, a robust version of the repeated-measures ANOVA and a robust post hoc test were performed, both using a significance level of *p* < 0.05. All the statistical analyses were carried out using R statistical computing software (version 4.3.2) and RStudio (version 2023.06.0+421 “Mountain Hydrangea”) for Windows.

## 3. Results

### 3.1. Spatiotemporal Parameters

The results of the descriptive statistics (mean, standard deviation, minimum, and maximum) for the spatiotemporal parameters are shown in Table A1. The repeated-measures ANOVA revealed that all the spatiotemporal parameters differed significantly, with large effect sizes, between FW and both walker-assisted tasks (AW and VRAW) but not between AW and VRAW, as shown in Table A2.

AW and VRAW, compared to FW, showed a reduction in stride length, an increase in the number of strides, a decrease in gait speed, an increase in cadence, and an increase in stance and swing time, as well as the time to complete the 10 MWT. This information is shown in Figure 3.

### 3.2. Hip Joint

The descriptive statistics for the hip joint parameters are presented in Table A3. There were no significant differences between the right and left parameters, indicating that both legs exhibited similar kinematic patterns at the hip joint.

In the sagittal plane, the repeated-measures ANOVA revealed significant differences between the FW and AW, as well as between the FW and VRAW for all the sagittal hip parameters for both legs, with large effect sizes (Cohen’s d > 0.8), as shown in Table A4. The sagittal parameters were similar between the AW and VRAW. In the coronal plane, the parameters H8 and H9 of the left leg showed significant differences between the FW and AW with large effect sizes, but these variables were similar between the FW and VRAW. In the transverse plane, the H12 parameter was significantly different between the FW and both walker-assisted tasks (AW and VRAW) in both legs, with large effect sizes.

Compared to the FW parameters, the hip kinematics of AW and VRAW in the sagittal plane were characterized by increased flexion at the heel strike (H1), increased flexion at the loading response (H2), lower extension during the stance phase (H3), increased flexion at toe-off (H4), and increased maximum flexion during the swing phase (H5). Conversely, the total sagittal plane range of motion (H6) was lower in both walker-assisted gaits than in FW, as shown in Figure 4. In the transverse plane, AW and VRAW only showed a statistically significant difference in H12 of the hip, presenting greater hip external rotation in the swing phase. In the coronal plane, no significant differences were found.

### 3.3. Knee Joint

Table A5 shows the descriptive statistics of the knee joint parameters. Unlike the hip joint, the participants exhibited significant differences between the right and left legs for some parameters across the tasks. For FW, the differences were in K7 (mean difference = 1.18° and effect size = 0.60), K9 (mean difference = 1.27° and effect size = 0.52), and K10 (mean difference = 1.37° and effect size = 0.52). For AW, the differences were in K5 (mean difference = 2.56° and effect size = 0.52) and K6 (mean difference = 2.53° and effect size = 0.58). VRAW showed differences in the same parameters as AW, with mean differences of 2.41° (effect size = 0.55) and 2.53° (effect size = 0.59), respectively. Despite these differences, none had large effect sizes.

The results of the repeated-measures ANOVA for the sagittal and transverse parameters of the knee joint showed significant differences for both legs in almost all the parameters (except K11 in the right leg) between the FW and both walker-assisted tasks (AW and VRAW), with large effect sizes for almost all the parameters, except K3, as shown in Table A6. No differences were found between the AW and VRAW. In the coronal plane, no significant differences were observed between any gait tasks.

In summary, AW and VRAW, compared to FW, exhibited increased flexion in the sagittal plane at the heel strike (K1), increased flexion at the loading response (K2), reduced maximum extension during the stance phase (K3), reduced flexion at toe-off (K4), lower maximum flexion during the swing phase (K5), and an overall reduction in the sagittal plane excursion (K6). In the transverse plane, the knee joint exhibited lower internal rotation in the stance phase (K11), greater external rotation in the swing phase (K12), and lower total plane excursion (K10), as shown in Figure 5. No significant differences were found in the coronal plane.

### 3.4. Ankle Joint

For the ankle joint, the descriptive statistics parameters are shown in Table A7. Both legs exhibited similar kinematic patterns at the ankle joint, except for the A3 parameter in FW, which presented a significant difference (mean difference = 1.07° and effect size = 0.83).

Similar to the hip and knee joints, the repeated-measures ANOVA revealed significant differences between the FW and both walker-assisted tasks (AW and VRAW) for the sagittal plane parameters, as shown in Table A8. However, for the ankle, no significant differences were found in the A3 parameter in any gait tasks, and significant differences in the A4 parameter were found only between the FW and AW in the left leg. No significant differences were found between the AW and VRAW. In the coronal plane, no significant differences were observed between any gait tasks.

The AW and VRAW tasks in the sagittal plane of the ankle parameters were characterized by increased dorsiflexion at the heel strike (A1), reduced maximum plantar flexion at the loading response (A2), increased dorsiflexion at toe-off (A4), increased dorsiflexion in the swing phase (A5), and an overall reduction in the sagittal plane excursion (A6), as shown in Figure 6. The A3 parameter showed no differences across the tasks, meaning the maximum dorsiflexion in the stance phase was similar for FW, AW, and VRAW. No significant differences were found in the coronal plane.

### 3.5. SEQ

The score obtained from the SEQ after the VRAW day was 57.7 ± 8.4. This score indicates that the VR environment was suitable for a VR rehabilitation system. In general, all the volunteers reported being comfortable and having fun during the VRAW. From the questions designed to detect frequent issues from VR applications, none of the volunteers reported discomfort, dizziness, nausea, or eye discomfort, and only one volunteer reported being neutral about being confused or disoriented during the task. All the volunteers expressed interest in continuing to use the VR system during an eventual rehabilitation program.

## 4. Discussion

This paper investigated the kinematic patterns of the lower limb joints of elderly subjects during three gait tasks at self-selected speeds: FW, AW, and VRAW. The main goal of this paper was to evaluate whether the differences between FW and AW are comparable to those between FW and VRAW and to analyze how AW and VRAW differ from each other to assess the effects of VR on gait-assisted tasks.

The main results highlight that both walker-assisted gait tasks (AW and VRAW) produced significantly different kinematic patterns of joint motion compared to FW, particularly in the sagittal plane. However, the joint kinematics between AW and VRAW were similar in both legs. The results suggest that the UFES vWalker was the major factor influencing the kinematic changes, not the VR environment. The combination of overground assistive devices, especially SW, gait analysis, and VR for rehabilitation is scarce in the literature, and the results of this study may provide an initial background for future developments in this field.

During the AW and VRAW tasks, the participants took more time to complete the 10 m walk, with a reduced stride length and an increased number of steps. The slower gait pattern featured more than double the stance time and 20% more swing time compared to FW. Although the participants had their body weight partially supported by the walker structure, the need to drive the device forward may have influenced these spatiotemporal characteristics. Previous studies that investigated the biomechanical properties of walker models have also provided evidence that slower walking is a major feature of this gait assistance method [34,53,54]. However, such features are expected in clinical rehabilitation scenarios, where more controlled walking patterns are required [34].

In the sagittal plane, the hip kinematics of the AW and VRAW tasks were characterized by increased flexion at the heel strike and loading response, reduced extension at the stance phase, increased flexion at toe-off, and maximum flexion at the swing phase. The total sagittal plane range of motion was reduced in the AW and VRAW gait compared to FW. The hip joints presented greater flexion parameters in AW and VRAW. The explanation for this feature lies in the structure of the UFES vWalker itself. The forearm supports were designed to lead the user into a forward flexion position of the trunk to aid in body weight support during walking, but this position influenced the entire hip gait cycle by keeping the hip in an increased flexion position. This scenario was not changed by the VRAW task.

However, it is important to highlight that the increase in hip flexion observed with the UFES vWalker may be related to an increased percentage of weight discharge in the device and greater activation of the lower limb muscles [55]. These properties have potential applications for the rehabilitation of musculoskeletal and neurological clinical conditions that require gait training with weight-bearing assistance. In particular, the walker can be used to address early gait recovery in various inpatient and outpatient settings, including hip and knee arthroplasties, ligament reconstructions of the knee, arthroscopic procedures of lower limb joints, bone fractures, stroke, and cardiopulmonary conditions [56,57,58,59]. Some conditions, however, will require specific adjustments to the walker to ensure patient safety. For example, patients who have undergone total hip arthroplasty must not walk with excessive hip flexion during early rehabilitation to prevent hip subluxation [58].

In the transverse plane, AW and VRAW also showed greater hip external rotation during the swing phase for both legs. Future studies must consider this change in transverse plane motion when studying clinical conditions where an increase in hip external rotation might represent some risk, such as hip instability [60].

The knee kinematics between the right and left legs revealed discrepancies during FW for some frontal and transverse parameters. Previous studies have shown that these parameters are highly variable and may reflect random sample variations [49,61]. The AW and VRAW gait tasks showed increased maximum knee flexion at the swing phase of the left leg and, consequently, the overall sagittal motion of this leg was slightly higher. This finding was unexpected, but the greater range observed for the K5 and K6 data suggests that this may reflect adaptations regarding walker maneuverability. Compared to FW, the AW and VRAW knee kinematics were initially characterized by increased knee flexion at the heel contact and loading response. During the stance phase, the knee did not reach maximum extension, and toe-off occurred with reduced knee flexion. The maximum knee flexion at the swing phase was reduced during the AW and VRAW tasks, and the overall sagittal motion was about 25% smaller than FW.

The AW and VRAW tasks showed less knee motion in all the parameters of the transverse plane, meaning that the participants developed a walking pattern with less rotation between the tibia and femur. This feature may have applications in clinical conditions that require supported gait training with local stability requirements, such as ligament reconstructions, incomplete spinal cord injury, or stroke [56,57].

For the ankle joint, changes in kinematics occurred in the sagittal plane. Compared to FW, the AW and VRAW tasks exhibited increased plantar flexion at the heel strike and loading response. Maximum dorsiflexion at the stance phase was not different from FW, but toe-off occurred much earlier, with increased dorsiflexion that was sustained throughout the swing phase. The total sagittal plane excursion of the ankle joint was reduced by approximately 38%.

The overall reduction in the joint range of motion observed during the AW and VRAW tasks may be associated with the forward trunk posture and the level of body weight support. These results confirm previous studies that showed that increased body weight assistance during walking significantly influenced spatiotemporal and kinematic parameters [54,55].

Additionally, future studies should address the influence of the forearm support’s height over the hip joint flexion posture and, consequently, on overall 3D joint kinematics and spatiotemporal parameters. In this paper, this height was adjusted according to the volunteer’s height. Another consideration for future studies is to adjust the admittance controller parameters for each volunteer, as emulating different haptic sensations may influence lower limb kinematics in the attempt to move an SW [34]. Conversely, if the objective of such studies is not a kinematic precision measurement but gait training, admittance control may be used as an accessory to lower limb strengthening protocols because it can be used to simulate dynamics behaviors according to the VR environment [32,35].

The VR scenario developed in this research consisted of a simple star-gathering game that encouraged the participants to walk in a straight line. The findings in this paper indicate that VR did not influence the walking kinematics parameters compared to a similar task performed without VR. Our results differ from the findings of [11,21,22,23], which reported changes in joint kinematics with VR gait. This discrepancy could be attributed to differences in VR applications across studies, which investigated overground free walking and treadmill gait. Furthermore, the posture adopted by the participants in the walker-assisted tasks was more stable, allowing for limited variations in the joint kinematics and spatiotemporal parameters. Future studies in the field of VR have enormous potential to develop gait rehabilitation strategies that mimic real-world challenges in clinical scenarios, which are typically found in the elderly population, potentially contributing to treatment quality and adherence.

During the experiments, some volunteers spontaneously shared their experiences while performing the tasks. The participants expressed enthusiasm about using the UFES vWalker, both with and without the VR headset. This excitement likely stemmed from their unfamiliarity with robotics and virtual reality technology. The level of enthusiasm appeared to increase when using the VR headset, as the participants were positively surprised by the virtual elements, such as the stars they collected and the feedback interface. Some volunteers were even surprised by the distance they had covered once they removed the VR headset. A common suggestion from the participants was to improve the comfort of the UFES vWalker’s forearm support, which we plan to address in the current iteration of our system.

Finally, the mean scores obtained from the SEQ indicate that the VR game was comfortable, and the participants did not report adverse effects. In this way, our results provide evidence that the data from the VRAW group were not influenced by cybersickness related to immersive VR environments. The factors that may mitigate adverse effects in the VRAW include the support offered by the UFES vWalker during the tasks, providing more stability and security for the volunteers during the walk. Additionally, the synchronized movement between the VR and the physical world, where every displacement by the volunteers and the UFES vWalker was reproduced in VR, likely contributed to these findings. This synchronization does not happen in VR treadmills, for example. Furthermore, the straight-line walk reduced the head’s range of motion, which may help to avoid cybersickness outcomes.

The use of the straight-line walk was made to establish a baseline for comparison between the FW, AW, and VRAW tasks. With this path, we could more confidently attribute any observed difference in gait kinematics to the presence of VR itself rather than to specific features of a complex virtual environment, such as turns or obstacles. Additionally, given our elderly participant group’s first contact with both a robotic walker and VR headset, a simple VR task helped minimize potential safety risks. Future research should explore different walking paths, such as incorporating curves, to assess how variations in head movement or motion might affect cybersickness symptoms.

## 5. Conclusions

This paper investigated the effects on gait kinematics of fourteen elderly participants in assisted locomotion within a VR scenario across three experimental tasks: normal free walking (FW), smart walker-assisted gait (AW), and smart walker-assisted gait with VR assistance (VRAW). The FW task was designed to capture each individual’s standard walking patterns, which were then compared with the patterns observed in the AW and VRAW tasks to assess if the changes from FW to AW were similar to those from FW to VRAW. After the VRAW task, the volunteers responded to the SEQ to evaluate the VR rehabilitation environment.

The main results highlight that both walker-assisted gait tasks (AW and VRAW) produced significantly different kinematic patterns of the spatiotemporal parameters and hip, knee, and ankle joint motions compared to FW, particularly in the sagittal plane. However, the joint kinematics between the AW and VRAW were similar in both legs, suggesting that VR did not influence SW gait kinematics. Both AW and VRAW were characterized by a reduction in the range of motion of the hip, knee, and ankle in the sagittal plane; a reduced stride length and gait speed; and an increased stride number, cadence, and time to complete the 10 MWT. The posture adopted by the participants in our SW model was very stable, allowing for very few variations in joint kinematics and spatiotemporal parameters. The scores from the SEQ suggest high usability with almost no side effects.

The similar gait parameters of AW and VRAW, combined with the high acceptance of the SEQ, indicate that VR environments could be a promising tool during rehabilitation programs with SW. The SW’s advanced features could be combined with the motivation and engagement of VR applications for the development of tasks with different cognitive and physical demands.

However, implementing the UFES vWalker in clinical settings presents challenges, including the need for prior training for both healthcare professionals and patients. Additionally, clinical facilities may require adaptations to accommodate the UFES vWalker and ensure its proper functioning.

Our research group is already developing some tools to overcome these challenges. Healthcare professionals will receive training with our research group to explain how to operate the UFES vWalker and select and customize rehabilitation scenarios in the VR headset. For patients, we are developing explainable user interfaces designed to provide instructions on safe navigation, moving forward, making turns, and avoiding both real and virtual obstacles in the VR headset. We are also focused on developing cloud-based systems for VR. Because VR may require significant computational power to render virtual obstacles and virtualize physical environments, cloud-based VR systems could facilitate remote data processing, enabling clinical facilities to only need the UFES vWalker and the VR headset while the cloud handles all data processing and provides remote support.

These tools will enable future research to conduct long-term clinical trials and develop interactive VR environments tailored to the specific needs of patients undergoing gait rehabilitation. This includes adjusting the height of the UFES vWalker according to individual patient requirements. By customizing both the walker’s design and the VR environment, we can create more effective and personalized gait rehabilitation protocols.

## Figures and Tables

**Figure 1 sensors-24-05534-f001:**
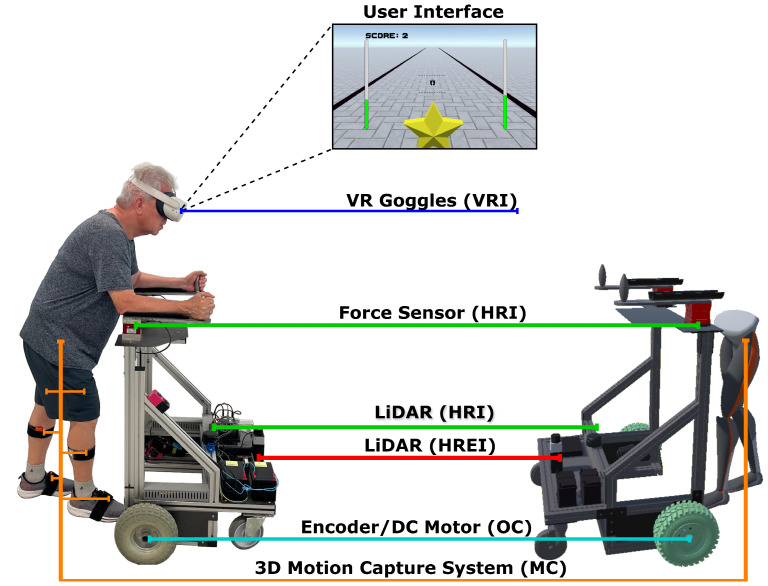
A participant from the experiment using the UFES vWalker and the five subsystems: Odometry and Control (OC), Human–Robot-Environment Interaction (HREI), Human–Robot Interaction (HRI), Motion Capture (MC), and Virtual Reality Integration (VRI).

**Figure 2 sensors-24-05534-f002:**
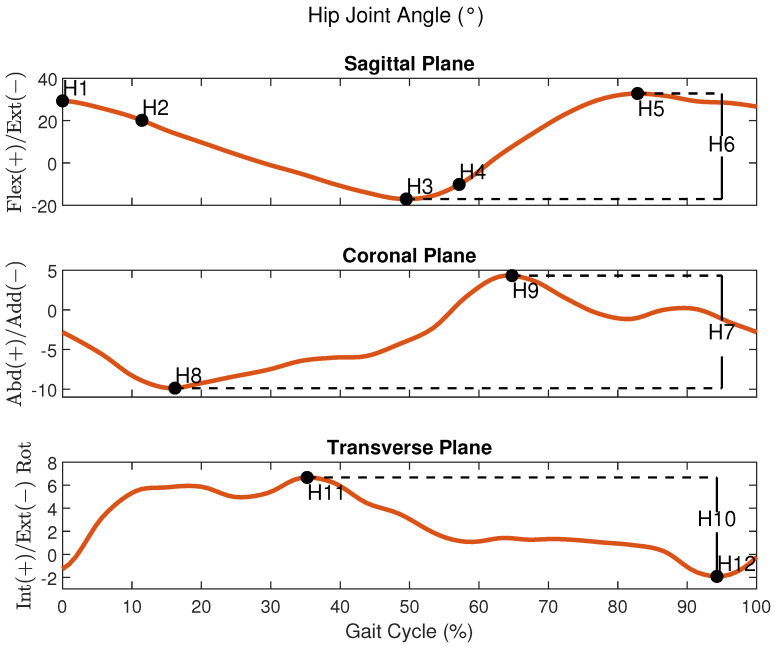
An example of the joint angles of the hip, knee, and ankle respectively during a gait cycle from a participant in the FW task.

**Figure 3 sensors-24-05534-f003:**
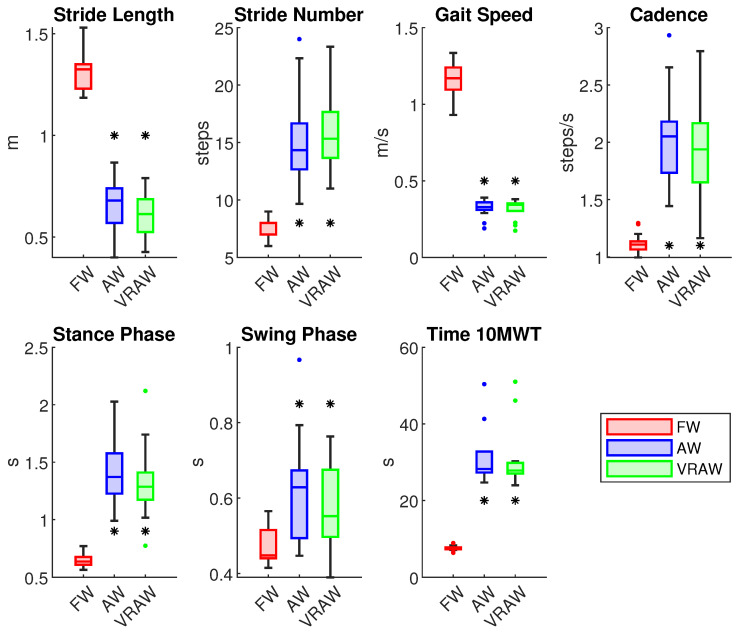
The boxplot of all the spatiotemporal parameters divided by the FW, AW and VRAW tasks. The boxplots for AW and VRAW, marked with an asterisk (*), indicate that statistically significant differences were found between these tasks and FW.

**Figure 4 sensors-24-05534-f004:**
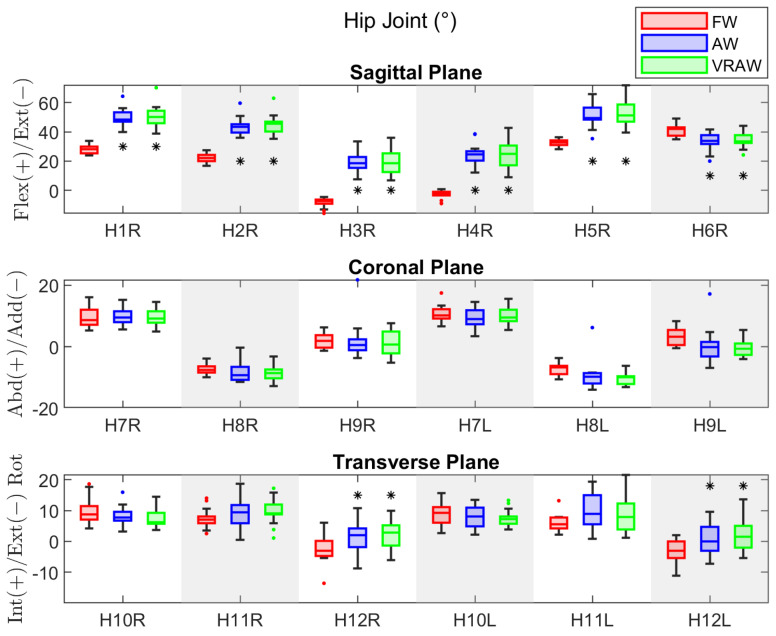
The boxplot of the kinematic parameters of the right hip joint in the sagittal plane, and both sides in the coronal and transverse planes, divided by the FW, AW, and VRAW tasks. The boxplots for AW and VRAW, marked with an asterisk (*), indicate that statistically significant differences were found between these tasks and FW.

**Figure 5 sensors-24-05534-f005:**
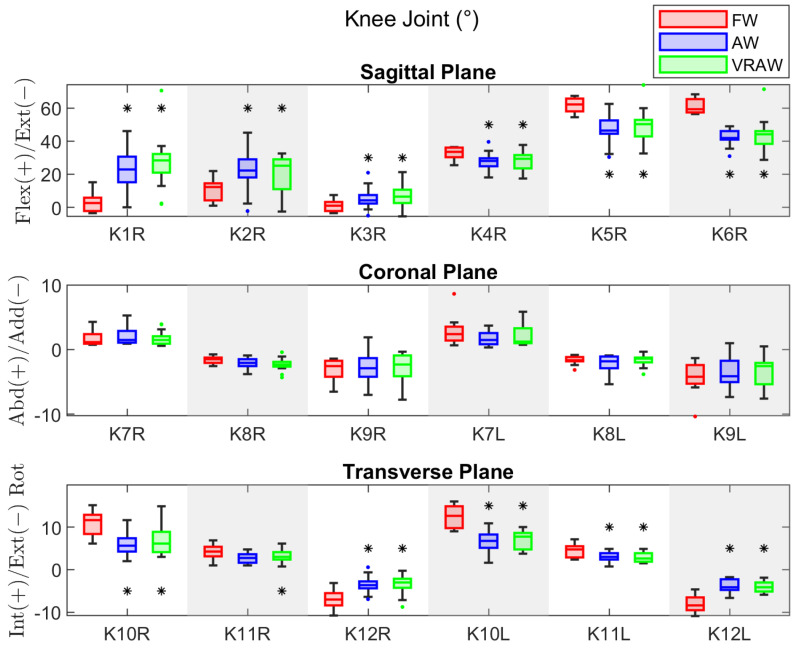
The boxplot of the kinematic parameters of the right knee joint in the sagittal plane, and both sides in the coronal and transverse planes, divided by the FW, AW, and VRAW tasks. The boxplots for AW and VRAW, marked with an asterisk (*), indicate that statistically significant differences were found between these tasks and FW.

**Figure 6 sensors-24-05534-f006:**
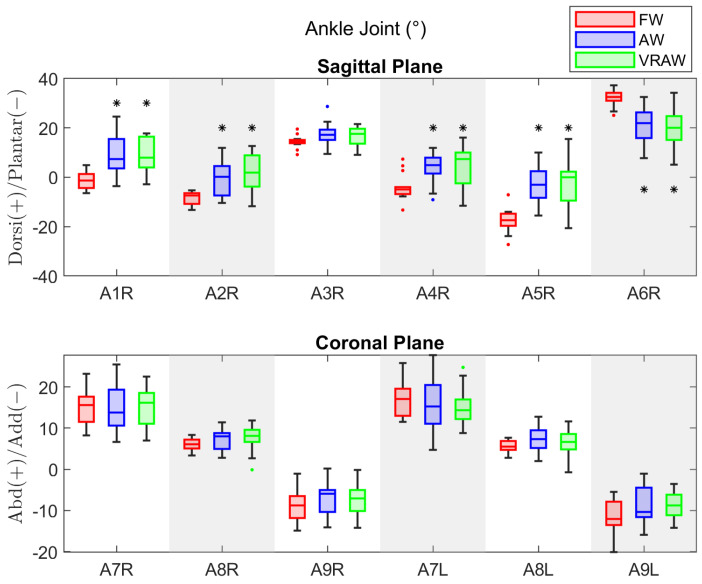
The boxplot of the kinematic parameters of the right ankle joint in the sagittal plane, and both sides in the coronal plane, divided by the FW, AW, and VRAW tasks. The boxplots for AW and VRAW, marked with an asterisk (*), indicate that statistically significant differences were found between these tasks and FW.

**Table 1 sensors-24-05534-t001:** The parameters of the lower limb joint angles of the hip, knee, and ankle analyzed.

Hip Joint	Knee Joint	Ankle Joint
H1: flexion at heel strike.	K1: flexion at heel strike.	A1: dorsiflexion at heel strike.
H2: max. flexion at load. response.	K2: max. flexion at load. response.	A2: max. plantar dorsiflex. at load. response.
H3: max. extension in stance phase.	K3: max. extension in stance phase.	A3: max. dorsiflexion in stance phase.
H4: flexion at toe-off.	K4: flexion at toe-off.	A4: dorsiflexion at toe-off.
H5: max. flexion in swing phase.	K5: max. flexion in swing phase.	A5: max. dorsiflexion in swing phase.
H6: total sagittal plane excursion.	K6: total sagittal plane excursion.	A6: total sagittal plane excursion.
H7: total coronal plane excursion.	K7: total coronal plane excursion.	A7: total coronal plane excursion.
H8: max. adduction in stance phase.	K8: max. adduction in stance phase.	A8: max. abduction in stance phase.
H9: max. abduction in swing phase.	K9: max. abduction in swing phase.	A9: max. adduction in swing phase.
H10: total transverse plane excursion.	K10: total transverse plane excursion.	
H11: max. internal rot. in stance phase.	K11: max. internal rot. in stance phase.	
H12: max. external rot. in swing.	K12: max. external rot. in swing phase.	

## Data Availability

The original contributions presented in the study are included in the article, further inquiries can be directed to the corresponding author.

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
