# Peer review of "Analysis of Gait Kinematics in Smart Walker-Assisted Locomotion in Immersive Virtual Reality Scenario"

_sensors, 2024, doi:10.3390/s24175534_

Round 1

Reviewer 1 Report

Comments and Suggestions for Authors

This research discusses a VR setting used within gait cinematics. The overall research is very good, but some aspects should be adapted a bit:

* I think it would be beneficial if you could add more concrete results within the abstract as well
* I would suggest to add bit more context to this sentence: Recently, technologies have been introduced into the biomechanical and rehabilitation fields and can be integrated into training (particularly gait) to improve outcomes, motivation, and 44treatment adherence [10]. (E.g. which technologies, more concrete)
* I would suggest to more details how and which people were selected. Please also explain „random“ in more detail
* Can you please also add more details, why Oculus Quest 2 was selected to be used?
* Please explain why the SEQ Questionnaire was used (and not any other)
* Since this might be a related topic: it might also be feasible to add something about gamification and serious gaming
* Could you also explain more about potential future work
* Overall the target group (and the focus of your SW) should be explained a bit more - e.g. rehabilitation was mentioned within the introduction but the evaluation was done with older people without a specific rehabilitation setting. I believe this research might be used in both settings, but this should be explained/discussed in more detail.
* In addition it would also be nice to add more feedback from the users (if there is more?) outside the questionnaire (e.g. are there any specific requirements from the target group)?

Comments on the Quality of English Language

Only some minor spelling issues

Reviewer 2 Report

Comments and Suggestions for Authors

The authors investigate the effects of smart walker (SW) assistance combined with virtual reality (VR) on the gait kinematics of elderly individuals. The study involves three tasks: free walking (FW), smart walker-assisted gait (AW), and smart walker-assisted gait combined with VR (VRAW). The researchers aimed to evaluate whether VR influences gait kinematics during SW-assisted walking and to assess the potential of VR as a tool in rehabilitation. The results show that while significant kinematic differences were observed between free walking and SW-assisted walking, VR did not significantly alter these kinematic patterns. The study concludes that VR can be effectively integrated into SW-assisted gait rehabilitation without compromising gait kinematics. However, after reading carefully, a few concerns are found and must be resolved before publication. 

1. The authors are suggested to use more recent works in the rehabilitation domain using VR in the introduction section. For example: "Integration of virtual reality and augmented reality in physical rehabilitation: a state-of-the-art review."

2. The study includes 14 elderly participants, but more details on the participant's health status, mobility levels, and previous experience with VR would be helpful. This information could help understand the variability in the kinematic data observed across different individuals.

3. The VR task involved a simple straight-line walking game. The authors should consider varying the complexity of the VR environment, such as incorporating turns or obstacles, to assess how these factors might influence gait kinematics and participant engagement.

4. The interpretation of hip, knee, and ankle joint angles data could be enhanced by relating the observed changes to specific clinical conditions or rehabilitation goals. For example, discussing how increased hip flexion during SW-assisted tasks might benefit or hinder a particular patient group would add practical relevance.

5. The design of the UFES vWalker, particularly the forearm supports, appears to influence trunk and hip posture. Discussing potential modifications to the walker design that could minimize unwanted postural changes would be valuable, especially if the goal is to maintain more natural gait patterns during rehabilitation.

6. The manuscript mentions repeated measures ANOVA and other statistical tests. Providing more details on how violations were addressed (e.g., Greenhouse-Geisser correction) would clarify the robustness of the findings.

7. In Figures 3 through 6, it would be helpful to provide more context in captions or annotations highlighting key findings or significant differences in the data.

8. The authors are suggested to discuss the practical implications of integrating VR with SW in rehabilitation settings. For instance, how might this technology be implemented in clinics or homes, and what training might be required for therapists and patients?

9. The conclusion suggests the potential of VR in gait rehabilitation, but further research is needed. Expanding on specific areas for future work, such as long-term clinical trials or developing adaptive VR environments tailored to individual rehabilitation needs, would strengthen the overall conclusion.

Round 2

Reviewer 2 Report

Comments and Suggestions for Authors

The manuscript may now be accepted as the authors have addressed all the concerns raised by the reviewer.